# Accelerated Early Progression of Amyotrophic Lateral Sclerosis over the COVID-19 Pandemic

**DOI:** 10.3390/brainsci11101291

**Published:** 2021-09-29

**Authors:** Fabiola De Marchi, Chiara Gallo, Maria Francesca Sarnelli, Ilaria De Marchi, Massimo Saraceno, Roberto Cantello, Letizia Mazzini

**Affiliations:** Department of Neurology and ALS Centre, Traslational Medicine, University of Piemonte Orientale, Maggiore della Carità Hospital, 28100 Novara, Italy; 20032512@studenti.uniupo.it (C.G.); mf.sarnelli@tiscali.it (M.F.S.); ilaria.demarchi1994@gmail.com (I.D.M.); 20031957@studenti.uniupo.it (M.S.); roberto.cantello@med.uniupo.it (R.C.); letizia.mazzini@uniupo.it (L.M.)

**Keywords:** amyotrophic lateral sclerosis, motoneuron disease, COVID-19, pandemic, progression, ALSFRS-R

## Abstract

During the COVID-19 pandemic and the related lockdowns, outpatient follow-up visits for patients with chronic neurological diseases have been suspended. Managing people affected by amyotrophic lateral sclerosis (ALS) has become highly complicated, leaving patients without the standard multidisciplinary follow-up. This study aimed to analyze the impact of the COVID-19 lockdown on ALS disease progression. We compared the clinical data and progression in the first year following diagnosis for patients who received ALS diagnosis during 2020 (G20, N = 34), comparing it with a group of diagnosed in 2018 (G18, N = 31). Both groups received a comparable multidisciplinary model of care in our Tertiary Expert ALS Centre, Novara, Italy. The monthly rate of ALSFRS-R decline during the lockdown was significantly increased in G20 compared to G18 (1.52 ± 2.69 vs. 0.76 ± 0.56; *p*-value: 0.005). In G20, 47% required non-invasive ventilation (vs. 32% of G18). Similarly, in G20, 35% of patients died vs. 19% of patients in G18 (*p*-value: 0.01). All results were corrected for gender, age, site of onset, and diagnostic delay. Several factors can be implicated in making ALS more severe, with a faster progression, such as reduced medical evaluations and the possibility of therapeutic changes, social isolation, and rehabilitation therapy suspension.

## 1. Introduction

“The COVID-19 Lockdown”, the mandatory isolation related to the ongoing pandemic by SARS-CoV-2 infection, has been primarily adopted by the Italian government since March 2020 to contain the virus spread both nationally and internationally (https://www.gazzettaufficiale.it/eli/id/2020/02/23/20A01228/sg, accessed on 25 April 2021).

Many countries have taken drastic measures to slow down infection rates to protect people’s health, especially older people and those with chronic diseases, the so-called “frail patients”. Indeed, the higher mortality rate in this group was evident, according to a higher risk of developing severe health consequences from COVID-19 [1,2,3].

Although the lockdown on vulnerable populations was an essential step, some critical aspects of social isolation and primary service interruption have recently been highlighted [4,5]. Indeed, these aspects negatively reflect physical, cognitive, and mental health outcomes for people with chronic neurological conditions, who need extensive and continuous care, regular physical activity, and daily routine and balance [6].

In the context of amyotrophic lateral sclerosis (ALS), the lockdown in Italy, similar to other European countries, partially prevented the regular diagnostic and therapeutic process and multidisciplinary follow-up at specialized reference ALS centers. The management of these frail patients, affected by a rapidly progressive disabling disease, has become enormously complicated, especially for those with significant functional and respiratory impairment, with a high risk of SARS-CoV-2 infection and unfavorable outcome [7]. A European group [8] reported a faster monthly rate decline of the ALS Functional Rating Scale–Revised (ALSFRS-R) score during the lockdown than the pre-lockdown period. Several factors may be involved in this phenomenon, such as the psychological consequences of lockdown and the social isolation, added to the awareness of fatal diagnosis, the interruptions of physical activity, and the caregiver’s distress [9,10,11]. In addition, mild mood disorders such as anxiety, emotional instability, and depression can be present in people with ALS [12] and may have worsened during the lockdown because of the sense of uncertainty, loneliness, and pandemic-related concerns. Furthermore, the psycho-emotional equilibrium of these patients may decline because of the additional stressors on caregivers, since the role of caregivers in contributing to patient wellbeing is well established [13]. It is imperative to consider the possible diagnostic disease delay, deriving from the problematic access to primary and specialized care for the first evaluation and the fear of contagion related to hospital access.

Based on these premises, this study aimed to retrospectively describe the overall impact of the lockdown on ALS progression by analyzing patients who received a diagnosis of ALS during 2020 and investigating the possible changes of functional status, disease severity, and the required interventions (e.g., non-invasive ventilation start or gastrostomy tube placement), comparing the 2020 data with a control group of ALS patients who received the diagnosis in 2018.

## 2. Materials and Methods

### 2.1. Study Design

This study was a retrospective analysis of data obtained enrolling ALS patients meeting the revised El Escorial Criteria for defined and probable-laboratory supported ALS (all patients underwent electromyography for diagnosis) diagnosed in 2020 at the Tertiary ALS Centre, Maggiore della Carità Hospital, University of Piemonte Orientale, Novara, Italy. Some patients (see group detail) were excluded from the analyses because of missing information on clinical evolution, absence of complete follow-up, or juvenile, only clinically probable and atypical forms. For the comparison group, we collected data from patients with ALS diagnosed in 2018, using the same inclusion criteria. Both groups received a comparable multidisciplinary model of care in our Tertiary Expert ALS Centre. Alternatively, for 2020, if the patient had scheduled a visit during the pandemic peak period, this was postponed and patients were contacted by telemedicine.

In detail:(1)Group 2020 (G20): in the pandemic year, we diagnosed ALS in 38 patients and followed during the disease course (or censored date, 30 April 2021); 34 were included in the study, four excluded for missing data.(2)Group 2018 (G18): in 2018, we diagnosed ALS in 82 patients and followed during the disease course (or censored date, 30 April 2021): 31 were included in the study, 51 excluded for missing data (of these 51 patients, 27 were excluded as patients led to our center only for a second-opinion visit and/or for diagnosis confirmation from other Italian regions but not followed for whole disease course in our center).

### 2.2. Data Collection

For each patient, we included extensive clinical information: gender, age at onset, symptoms onset date, diagnosis date, phenotype (bulbar or spinal), mutational status for c9orf72, SOD1, TARDBP, and FUS, neuropsychological profile according to the revised ALS-frontotemporal spectrum disorder (ALS-FTD) Consensus Criteria, use of non-invasive or invasive ventilation based on EFNS guidelines [14] (and date of beginning), feeding tube placement (and date of placement), and survival time (time from symptoms onset and death or tracheotomy). Clinical data as ALS Functional Rating Scale–Revised score (ALSFRS-R), body mass index (BMI), and forced vital capacity (FVC%, expressed as a percentage of predicted values) were collected at diagnosis and for whole disease duration as part of the multidisciplinary care follow-up. In this analysis, we considered and compared only the data obtained monitoring the first year of disease (from diagnosis) for the two groups. The monthly ALSFRS-R progression rate was calculated using the delta formula: (ALSFRS-R at diagnosis–ALSFRS-R at follow-up)/(last date–first date). A similar analysis was conducted for delta FVC% progression and delta BMI. All scores included in this analysis were collected by a dedicated physician, in an in-person setting at least for the first and the last evaluation; some intermediate measures were obtained by telemonitoring.

For patients of G20, we analyzed the psychological assessment and support with the Hospital Anxiety and Depression Scale—HADS—and with the ALS Assessment Questionnaire (ALSAQ-40) collected during a psychological/neuropsychological evaluation during the standard multidisciplinary care. In addition, in this group, we investigated the patient’s opportunity to continue the physiotherapy rehabilitation treatment and the eventual initiation of antidepressant therapy.

### 2.3. Statistical Analyses

Descriptive statistics were used for demographic information. We first performed simple comparison analyses between the groups (G20-G18) for each of the included variables. Univariate analyses were completed using either a chi-square test (for categorical variables) or independent samples *t*-test (for continuous variables). We added a linear regression model for ALSFRS-R progression. The analyses were corrected for age, sex, phenotype, diagnostic delay, ALSFRS-R, FVC%, and BMI at baseline. Survival was calculated from diagnosis to death/tracheostomy or censoring date (30 April 2021), using Kaplan–Meier survival curves followed by log-rank test (chi-square values, *p*-value). All tests were assessed at an alpha of 0.05. All statistical procedures were carried out with SPSS Version 25.0 (SPSS Inc., Chicago, IL, USA).

### 2.4. Data Availability

The study was conducted in strict accordance with the principles of the Declaration of Helsinki. Prospective informed consent and ethical review and approval were waived due to the retrospective nature of the study and the use of pseudonymized data. No risks were expected for the subjects for the retrospective study design, and the results of the study did not impact the diagnosis, prognosis, or management of study participants. The datasets generated during the analysis of this study are available from the corresponding author on reasonable request.

## 3. Results

### 3.1. Patient Characteristics

A total of 65 patients with ALS followed up for the whole disease duration were included (data from 34 patients from 2020 and data from 31 patients from 2018). All participants’ demographic and clinical characteristics at the time of ALS diagnosis are presented in Table 1. At baseline, no differences are observed in clinical variables, such as age at onset, sex, onset region, and symptoms’ duration before diagnosis. At diagnosis, we did not observe significant differences in ALSFRS-R, FVC%, and BMI at baseline, though a lower score in each of these was found in G20 at diagnosis (Table 1).

### 3.2. ALS Disease Progression in the First Year of the Disease

During the first year of disease, we evaluated G20 ALS patients as an outpatient service for a mean of 3.76 visits/year. For the G18 group, we performed 4.77 visits/year. We observed a monthly ALSFRS-R progression of 1.52 (SD: 2.69) in G20 and 0.76 (SD: 0.56) in G18 (*p*-value = 0.005). Also in a linear regression model, the ALSFRS-R progression is related only to belonging to G20, with respect to G18. However, we observed also a slightly higher monthly decrement of FVC% in G20, without a statistically significant difference. We did not observe a significant difference in BMI between the two groups, although we did observe less weight loss in G20 than G18 (see Table 2).

For ventilatory support, in G20, 15/34 patients (44%) required non-invasive ventilation (NIV), and one of these underwent a tracheostomy. In G18, in the first year of disease, 10/31 patients (32%) started NIV and none had a tracheostomy in this analyzed period (Figure 1). For the G20 group, the mean time between diagnosis and NIV onset was 4.56 months (SD: 3.26); for G18, it was 5.8 months (SD: 2.30) (*p*-value > 0.05).

For nutrition, during the observational period (first year of disease), 8/34 patients (23.5%) required supplementary nutrition (3/8 with nasogastric tube feeding, and 5/8 had a gastrostomy tube placed) in G20; in G18, 7/31 patients (22.5%) had a gastrostomy tube placed (Figure 2). For the G20 group, the mean time between diagnosis and nutritional support onset was 5.75 months (SD: 4.30); for G18, it was 7.85 months (SD: 3.30) (*p*-value > 0.05).

### 3.3. Survival

In G20, 12/34 patients died (35%) (median time from symptom onset to death: 18 months, IQR 10.5–25.3), with a mean age of 71.25 (SD 10.18); in G18, 6/31 patients died (19%) (median time from symptom onset to death: 35 months, IQR: 25.5–47), with a mean age of 66.50 (SD: 13.54) (for percentage survival, chi-square test: 6.03, *p*-value 0.01). One patient of G20 died due to the COVID-19 infection. No other patients in these groups had the COVID-19 infection. In Figure 3, the Kaplan–Meier curve is shown for the first year of disease.

### 3.4. Psychological Assessment and Medication Use

Anxiety and depression HADS scores and ALSAQ-40 scores for G20 are reported in Table 3. For G20, medication use for anxiety and depression management was collected, and 13/34 patients (38%) started at least one antidepressant in the first year of disease, coinciding with the COVID-19 pandemic. For G18, 11/31 (35%) started antidepressant drugs. Due to the lockdown limitations, only 15 of 34 (44%) of G20 were able to carry out the physiotherapy program.

## 4. Discussion

In this study, we highlighted a faster ALSFRS-R worsening in patients who received a diagnosis of ALS during the pandemic compared to a group of patients with ALS diagnosed in 2018 as a standard reference group. Both groups received an ALS diagnosis at our Tertiary Center and were followed up for all disease courses (death or censored date) with the same multidisciplinary team. Of these patients, we investigated the eventual change of functional status, disease severity—including survival—and the need for intervention (e.g., non-invasive ventilation start or gastrostomy tube placement). The limited analysis of the first year of disease in both groups is related to the need to make the two groups homogeneous and avoid bias for the well-known possibility of disease plateau over the course [15].

In 2020, compared to 2018, roughly half of the patients presented to our ALS Tertiary Center for a diagnostic evaluation. The greater number of patients who received a diagnosis of ALS at our center in 2018 is due to the fact that our center is a national Tertiary Center and, in non-pandemic periods, many patients come from other Italian locations for diagnosis/diagnostic confirmation. However, these patients then, for geographic reasons, are followed during the course of the disease by centers closer to their home (not included in the analysis). In 2020, the patients who went to the Center were almost all residents in the Piedmont region (the same region of the Center), and no patients from other Italian regions were presented. However, the number of patients led to our Center from the Piedmont region in 2020 was the same as in 2018, reducing the probability of a referral bias in the groups’ composition. Similarly, we excluded that patients with less significant or more slowly progressive disease may choose not to delay seeing a doctor for diagnosis, which would mean they were not included in the G20 cohort, and that those patients who were very symptomatic from their disease or with rapidly progressing symptoms came to the Center because the baseline features were similar between the two groups. We believed in a reduction of hospital accesses for diagnostic confirmations from other Italian regions.

Independent from the tragic health situation, it is important to underline that all patients were followed up with the same multidisciplinary approach. The pandemic influence occurred on the follow-up visit number, which had to occur with longer latency for the imposed access limitation.

We demonstrated that G20 had a significantly faster functional decline in the ALSFRS-R score and a shorter survival time, with a higher death probability in the first disease year, compared to G18.

As reported in Table 1, the demographic and phenotypic patient features of the two groups at baseline did not differ from each other. In particular, despite the pandemic period and the evident reduction in hospital admissions for outpatient visits, no greater diagnostic delay was observed in the year 2020 for ALS patients, though, as discussed above, a significantly lower number of patients were admitted for the second expert opinion. Similarly, at baseline, the ALSFRS-R score was slightly lower than that of G18 but not statistically significant.

During the follow-up, we observed in G20 a significantly faster decrement in the monthly ALSFRS-R score (*p*-value: 0.005), which turned out to be double that of the G18 group. Not surprisingly, albeit statistically not significant, we also observed a higher monthly progression of FVC% and BMI. In addition, a higher number of patients had to start NIV due to respiratory function worsening. A similar number of patients required artificial nutritional supplementation, but we placed nasogastric tubes in almost 40% of cases. This change is explained both by the difficulty of admitting patients for the PEG positioning (compared to the nasogastric tube that could be performed on an outpatient visit) and by a worse respiratory function of the patients, which made the PEG positioning riskier for acute complications.

Faster disease progression with a rapid progression in the functional scale added to the need for more devices (e.g., artificial nutrition, NIV) is reflected in more deaths—and consequently, in a shorter survival time—in the first year of disease in the G20 group than in the G18 group. However, only one death in G20 is linked to COVID-19, and thus we did not consider the infection a survival bias. In addition, no other patients included in the analysis had a COVID-19 infection in the analyzed period, which may have accelerated the disease course.

To evaluate which factors could have influenced a more evident worsening in 2020 than in previous years, we administered—to the G20 patients still living at the censored date—anxiety and depression rating scales. Quite unexpectedly, the HADS and the ALSAQ40 revealed only mild levels of mood disorders. However, contrast results on this topic were already obtained in two other different groups of ALS Italian patients [16,17]. This phenomenon is clinically confirmed by a similar percentage in the two groups of antidepressant drugs prescribed in the first year of the disease.

More significant could be the reduced access to outpatient physiotherapy sections (both motor and respiratory) and rehabilitation hospitalizations during the lockdown, as already hypothesized [18]: indeed, we advised these physical treatments in almost the totality of patients, but less than half were able to follow it this year.

Only one similar study on the French ALS population was recently published [8], where an intra-rate analysis described a patient worsening during the COVID-19 pandemic compared to the previous period. Both studies agree on a more rapid worsening of patients during the pandemic evaluated with the ALSFRS-R score, albeit with two different analyses (an intra-group study design in the French study and an inter-group analysis in our study). Similar results were also obtained for the BMI trend. These results, obtained from the analysis of patients in different phases of the disease [8] and from patients in the first year of the disease, lead us to hypothesize that the transient disease-related fluctuations do not play a predominant role in clinical worsening during the lockdown. Furthermore, our study also analyzed the survival data, with solid results and significant impact in G20.

Our data do not have the potentiality to identify the precise cause of faster disease progression and related shorter survival and higher risk of death in our ALS cohort, but we can believe in a decisive role of the lockdown and the related restrictions, in terms of limited access in the Hospital (due to both the Italian government rule and patients’ contagion fear), of reduction in physiotherapy, social isolation, caregiver burden, and mood disorders.

The main limitations in the study are the small number of included patients, which limits some potential statistical significance, and short follow-up but related to the (still ongoing) pandemic. In addition, a further bias can be related to the lockdown: indeed, it could be that, during the pandemic, only patients with more rapid progression and a less favorable clinical course sought medical care, in contrast to a possible more heterogeneous group (with a higher percentage of slow progression) who sought care during a “standard” period. In addition, we did not collect evaluations for mood disorders in G18 and patients who died before the censored date.

## 5. Conclusions

We believe that the analysis of this period provides a unique opportunity for clinicians, researchers, and health authorities to observe how the lockdown influenced ALS progression, prevent the consequences of eventual future sanitary restrictions and adapt them to the patients’ needs, and carry out good clinical practice.

## Figures and Tables

**Figure 1 brainsci-11-01291-f001:**
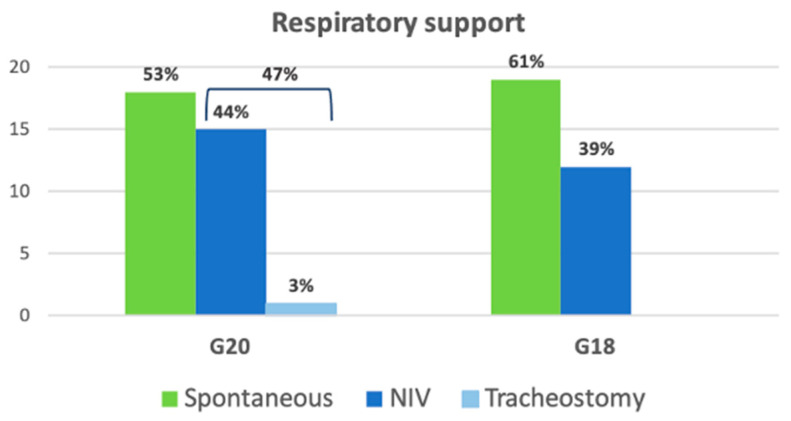
Respiratory support for ALS patients in G20 and G18 groups at one-year follow-up. G20: ALS patients with diagnosis in 2020; G18: ALS patients with diagnosis in 2018; NIV: non-invasive ventilation.

**Figure 2 brainsci-11-01291-f002:**
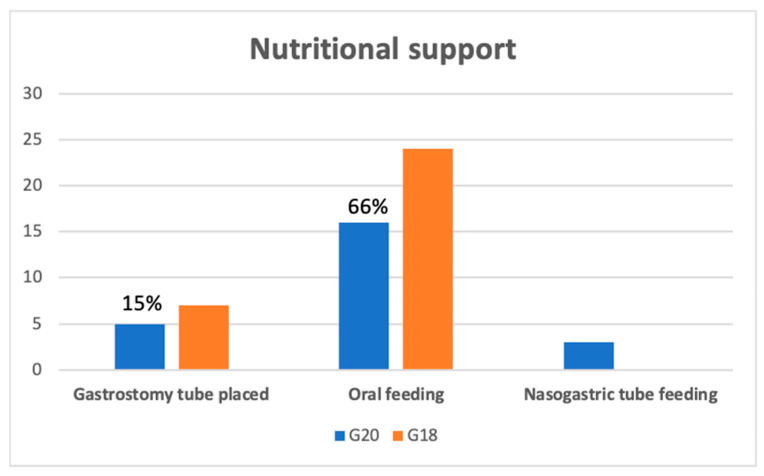
Nutritional support for ALS patients in G20 and G18 groups at one-year follow-up. G20: ALS patients with diagnosis in 2020; G18: ALS patients with diagnosis in 2018.

**Figure 3 brainsci-11-01291-f003:**
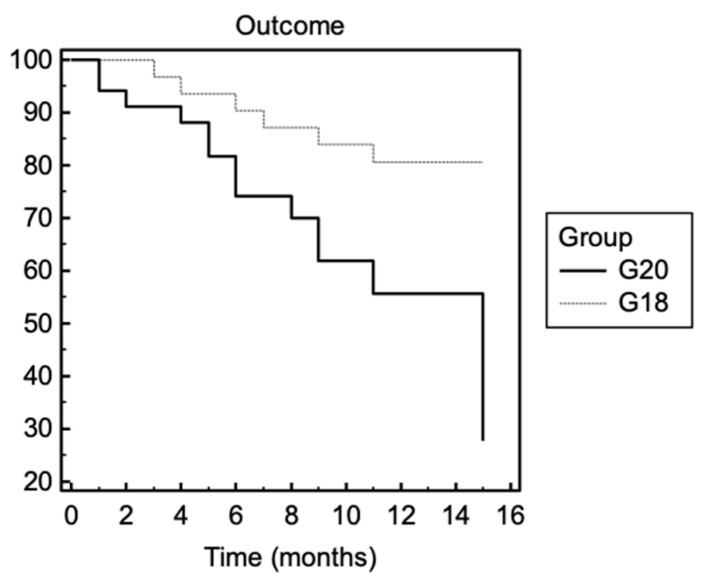
Survival in ALS patients in G20 and G18 groups for the first year of disease. G20: ALS patients with diagnosis in 2020; G18: ALS patients with diagnosis in 2018. Survival means death or tracheostomy. Chi-square test: 6.03, *p*-value = 0.01.

**Table 1 brainsci-11-01291-t001:** Demographic and clinical characteristics of the study participants. G20: ALS patients with diagnosis in 2020; G18: ALS patients with diagnosis in 2018. N: number; SD: standard deviation; IR: interquartile range; ALSFRS-R: Amyotrophic Lateral Sclerosis Functional Rating Scale–Revised; FVC: forced vital capacity; BMI: body mass index. Values are reported as mean (standard deviation, SD) or median (interquartile range, IQR).

	G20 (n = 34)	G18 (n = 31)	*p*-Value
Age (years) at onset (SD)	66.23 (9.90)	62.83 (13.43)	>0.05
Male/Female (%)	19 (56%)/ 15 (44%)	15 (48%)/ 16 (52%)	>0.05
Months from onset to diagnosis (IQR)	9.00 (6.24–12.76)	10.50 (6.00–12.00)	>0.05
Phenotype (%): - spinal - bulbar	24 (71%) 10 (29%)	19 (61%) 12 (39%)	>0.05
Cognition (%): - normal - impaired	21 (62%) 13 (38%)	23 (74%) 8 (26%)	>0.05
Gene mutations (%): - negative - C9Orf72	31 (91%) 3 (9%)	29 (94%) 2 (6%)	>0.05
ALSFRS-R at baseline (SD)	38.51 (5.34)	39.03 (5.35)	>0.05
FVC% at baseline (SD)	75.41 (22.27)	84.00 (16.63)	>0.05
BMI at baseline (SD)	23.90 (4.48)	25.35 (5.36)	>0.05

**Table 2 brainsci-11-01291-t002:** ALSFRS-R, FVC%, and BMI monthly progression in the first year of disease. G20: ALS patients with diagnosis in 2020; G18: ALS patients with diagnosis in 2018. ALSFRS-R: Amyotrophic Lateral Sclerosis Functional Rating Scale–Revised; FVC: forced vital capacity; BMI: body mass index; n.s.: not significant. Δ: monthly progression. All values are reported as mean and standard deviation. In bold: statistically significant, considering *p*-values < 0.05.

Outcome	G20	G18	*p*-Value
ΔALSFRS-R	1.52 (2.69)	0.76 (0.56)	**0.005**
ΔFVC%	1.97 (1.50)	1.74 (2.35)	>0.05
ΔBMI	0.25 (0.62)	0.05 (0.20)	>0.05

**Table 3 brainsci-11-01291-t003:** HADS and ALSAQ-40 mean score for G20. All values are reported as mean and standard deviation.

Scale	Score	Meaning
HADS–depression (mean, SD)	6.86 (5.74)	Absence
HADS–anxiety (mean, SD)	7.2 (5.95)	Mild level
ALSAQ-40–physical mobility (mean, SD)	40.13 (29.66)	Problems sometimes
ALSAQ-40–independence (mean, SD)	37.63 (27.48)	Problems rarely
ALSAQ-40–eating (mean, SD)	39.00 (44.01)	Problems rarely
ALSAQ-40–communication (mean, SD)	31.84 (41.65)	Problems rarely

## Data Availability

The datasets generated during the analysis of this study are available from the corresponding author on reasonable request.

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
