# Peer review of "Accelerated Early Progression of Amyotrophic Lateral Sclerosis over the COVID-19 Pandemic"

_brainsci, 2021, doi:10.3390/brainsci11101291_

Round 1
Reviewer 1 Report
The study analyses the impact of the COVI19-lockdown on ALS disease progression during the first year that follows diagnosis, by comparing two groups of patients followed in Italy, in the same ALS center, prior to (diagnosis in 2018) and during (diagnosis in 2020) COVID pandemic. The authors demonstrate that, in comparison with the 2018 patients, the 2020 patients experienced a faster progression of their symptoms, assessed by ALSFRS-R, accompanied by an increased proportion of patients requiring non-invasive ventilation, and an increased proportion of patients who succombed from their disease during this first year. The study contributes to emphasize the importance of the frequency of medical evaluations that allows a fast adaptation of the treatment and rehabilitation therapy, and the deleterious impact of social isolation on ALS patients.
The article is timely, well written and clear. However, it seems that the data could be further interrogated in an attempt to better understand the reasons for a faster progression of the 2020 patients.
1- The methods indicate that 38 patients were diagnosed in 2018 (and 34 included for the study), versus 82 in 2020 (and 31 included in the study). The difference is important, and the reason why many more patients were diagnosed in 2020 should be addressed in the first place. There is also a discrepancy with the discussion (lines 203-204) that reads "However, the number of patients led to our Center from the Piedmont region in 2020 was the same as in 2018, reducing the probability of a referral bias in the groups’ composition". These two related points have to be clarified.
2- While the initial data collected at time of diagnosis (ALSFRS, BMI, FVC%, month from onset to diagnosis) are not significantly different between the two groups, it appears clearly that overall the G20 group was already more affected at the beginning of the study than the G18 groups, and the absence of significativity may simply be a result of the relatively small size of the groups. The abstract (but not the methods) indicate that all data were corrected for age, gender, site of onset and diagnostic delay). It would be interesting to correct the data for initial ALSFRS, BMI and FVC% as well.
3- Data regarding nutrition should be presented in a figure similar to Figure 1 (Respiratory support), because they may illustrate that in 2020, patients were offered more rapidly a gastrotomy tube, and no nasogastric tube. Is this an evolution of the medical practice in this center, or a consequence of the lockdown ?
4- While these are somehow covered by the ALRFRS evaluation, the time between diagnosis and initiation 1) of respiratory support, 2) of supplementary nutrition and 3) of antidepressant should be indicated (and if possible illustrated), as an aditionnal indicator of disease progression and its consequences on the psycological state of the patients.
5- While the two groups had comparable average age at the time of initiation of the study, 2018 patients who died during the first year were on average younger than the 2020 patients. Is this difference significant ? If yes, could the authors comment in the discussion ?
6- The 2018 patients' group did not undergo psychological assessment, thus the two groups cannot be compared. In addition; less than 50% of the 2020 patients underwent psychological assessment. This is far too little to attempt a comparison with data from the litterature. While we aknowledge the attempt of the authors to assess this important aspect of the disease, we think that the data cannot be exploited and interpreted here, and should be removed. Accordingly, this part of the discussion should be removed as well (lines 243-254).
7- As the authors indicate in their introduction, the lockown "prevented the regular diagnostic and therapeutic process and multidisciplinary follow-up at specialized reference ALS centers". It would be particularly informative to compare between the two groups the number of medical visits the patients had during the first year of treatment that the study covers, and test whether it correlates with disease progression. If available similar correlation between disease progression and quantitative values reflecting when and how much the patients benefitted from physical therapy or other type of care would also be informative in the attempt to correlate disease progression with quanty and quality of medical visits and available therapies. While the point is raised in the discussion "The pandemic influence occurred on the follow-up visit number" (line 213-214), and "More significant could be the reduced access to outpatient physiotherapy sections (both motor and respiratory) and rehabilitation hospitalizations during the lockdown, as already hypothesized [15]: indeed, we advised these physical treatments in almost the totality of patients, but less than half were able to follow it this year" (lines 255-258), it should also be properly adressed in the results.
Minor point:
- line 88, "leaded" should be replaced by "led", or "addressed"
- - line 245, "he" should be replaced by "the"
Author Response
Response to reviewer 1:
We would like to thank the reviewer for careful and thorough reading our paper. Our response follows the reviewer comments, which are in are in italics.
Comment 1 - Reviewer #1: Content:
The methods indicate that 38 patients were diagnosed in 2018 (and 34 included for the study), versus 82 in 2020 (and 31 included in the study). The difference is important, and the reason why many more patients were diagnosed in 2020 should be addressed in the first place. There is also a discrepancy with the discussion (lines 203-204) that reads "However, the number of patients led to our Center from the Piedmont region in 2020 was the same as in 2018, reducing the probability of a referral bias in the groups’ composition". These two related points have to be clarified.
Reply to reviewer: thanks for the possibility to better clarify this point. There was probably an incomprehension because we included 34 patients in 2020 (diagnosed: 38) and 31 in 2018 (diagnosed: 82) (not as you wrote in the comment). The greater number of subjects who received a diagnosis of ALS at our center in 2018 is due to the fact that our center is a national center of excellence and, in non-pandemic periods, many patients come from other Italian locations for diagnosis / for diagnostic confirmation. However, these patients then, for geographic reasons, are followed during the course of the disease by centers closer to their home (and therefore, these patients were not included in the analysis -> hence the net reduction between patients evaluated and patient followed). Therefore, the number of patients treated in our center as follow-up visits did not change between 2018 and 2020.
Comment 2 - Reviewer #1: Content:
While the initial data collected at time of diagnosis (ALSFRS, BMI, FVC%, month from onset to diagnosis) are not significantly different between the two groups, it appears clearly that overall the G20 group was already more affected at the beginning of the study than the G18 groups, and the absence of significativity may simply be a result of the relatively small size of the groups. The abstract (but not the methods) indicate that all data were corrected for age, gender, site of onset and diagnostic delay). It would be interesting to correct the data for initial ALSFRS, BMI and FVC% as well.
Reply to reviewer: thanks for the possibility to improve this point. With a linear regression model, we evaluated what factors influenced the ALSFRS-R progression and only the belonging to a group (G20 and G18) is significant for progression (p-value 0.02). P-values are > 0.05 for age, sex, phenotype, FVC%, BMI and diagnostic delay.
Comment 3 - Reviewer #1: Content:
Data regarding nutrition should be presented in a figure similar to Figure 1 (Respiratory support), because they may illustrate that in 2020, patients were offered more rapidly a gastrotomy tube, and no nasogastric tube. Is this an evolution of the medical practice in this center, or a consequence of the lockdown ?
Reply to reviewer: thanks for this advice. Now we added Figure 2 with the nutritional support. The greater number of nasogastric tube placements is linked to the lower possibility of hospitalizing patients during the lockdown (see in Duscussion lines 230-240).
Comment 4 - Reviewer #1: Content:
While these are somehow covered by the ALRFRS evaluation, the time between diagnosis and initiation 1) of respiratory support, 2) of supplementary nutrition and 3) of antidepressant should be indicated (and if possible illustrated), as an aditionnal indicator of disease progression and its consequences on the psycological state of the patients.
Reply to reviewer: we calculated the time between:
- respiratory support onset: 4.56 months (SD 3.26) for the G20; 5.80 (SD 2.3) for the G18 (p-value > 0.05)
- supplementary nutrition onset: 5.75 months (SD 4.30) for the G20; 7.85 (SD 3.3) for the G18 (p-value > 0.05)
- antidepressant onset: 5.66 months (SD 4.76) for the G20; 7.01 (SD 3.3) for the G18 (p-value > 0.05)
Comment 5 - Reviewer #1: Content:
While the two groups had comparable average age at the time of initiation of the study, 2018 patients who died during the first year were on average younger than the 2020 patients. Is this difference significant? If yes, could the authors comment in the discussion?
Reply to reviewer: no, the p-value in 0.4 (probably related to the small sample) and so this difference is considered to be not statistically significant.
Comment 6 - Reviewer #1: Content:
The 2018 patients' group did not undergo psychological assessment; thus the two groups cannot be compared. In addition; less than 50% of the 2020 patients underwent psychological assessment. This is far too little to attempt a comparison with data from the literature. While we acknowledge the attempt of the authors to assess this important aspect of the disease, we think that the data cannot be exploited and interpreted here, and should be removed. Accordingly, this part of the discussion should be removed as well (lines 243-254).
Reply to reviewer: thanks for the suggestion. We decided to maintain this part in the results (according also with your comment n. 4) and we reduced the dedicated paragraph in the discussion.
Comment 7 - Reviewer #1: Content:
As the authors indicate in their introduction, the lockdown "prevented the regular diagnostic and therapeutic process and multidisciplinary follow-up at specialized reference ALS centers". It would be particularly informative to compare between the two groups the number of medical visits the patients had during the first year of treatment that the study covers, and test whether it correlates with disease progression. If available similar correlation between disease progression and quantitative values reflecting when and how much the patients benefitted from physical therapy or other type of care would also be informative in the attempt to correlate disease progression with quanty and quality of medical visits and available therapies. While the point is raised in the discussion "The pandemic influence occurred on the follow-up visit number" (line 213-214), and "More significant could be the reduced access to outpatient physiotherapy sections (both motor and respiratory) and rehabilitation hospitalizations during the totality of patients, but less than half were able to follow it this year" (lines 255-258), it should also be properly adressed in the results.
Reply to reviewer: as added in the main text, we evaluated G20 with a mean of 3.76 visits/year in the first year, compared to G18 group with a mean number of 4.77. So, for mandatory causes, we evaluated a bit less frequent G20 patients. In addition to this, we should consider the pandemic influence not only in the number of hospital follow-up visit but also in home treatment. This data is anamnestic collected through a questionnaire to patients but it is not possible to obtain the number of rehabilitation sessions at home in the two periods.
Reviewer 2 Report
The current manuscript aimed to retrospectively describe the overall impact of the lockdown on ALS progression, analyzing patients who received a diagnosis of ALS during 2020 and investigating the possible changes of functional status, disease severity, and the required interventions (e.g., non-invasive ventilation start or gastrostomy tube placement), comparing the 2020 data with a control group of ALS patients who received the diagnosis in 2018. This is a timely manuscript to determine the impact of COVID 19 on the ALS patients, however the following concerns needs to be addressed.
- How does the COVID-19 infection to the ALS individuals affected the progression of the disease?
- For ventilatory support, in G20, 15/34 patients (44%) required Non-Invasive Ventilation (NIV), and one of these underwent a tracheostomy. In G18, in the first year of disease, 10/31 patients (32%) started NIV and none had a tracheostomy in this analyzed period. How does COVID influences the rate of onset of clinical signs? Needs justification for this conclusion.
- Does ALS groups of G20 and G18 were taking any medications for depression before diagnosing the ALS itself? If yes, can you justify this?
Author Response
Comment 1 - Reviewer #2: Content:
How does the COVID-19 infection to the ALS individuals affected the progression of the disease?
Reply to reviewer: thanks for the possibility to improve our bibliography. We added three references that show how the prognosis of ALS patients with COVID-19 infection is worse compared to standard progression.
1.Li X, Bedlack R. COVID-19-accelerated disease progression in two patients with amyotrophic lateral sclerosis. Muscle Nerve. 2021 Sep;64(3):E13-E15. doi: 10.1002/mus.27351. Epub 2021 Jun 23. PMID: 34131925; PMCID: PMC8441768.
2.Digala LP, Prasanna S, Rao P, Govindarajan R, Qureshi AI. Impact of COVID- 19 Infection Among Hospi-talized Amyotrophic Lateral Sclerosis Patients. J Clin Neuromuscul Dis. 2021 Mar 1;22(3):180-181. doi: 10.1097/CND.0000000000000335.
3.Galea MD, Galea VP, Eberhart AC, et al. Infection rate, mortality and characteristics of veterans with amy-otrophic lateral sclerosis with COVID-19. Muscle Nerve. 2021;64(4):E18-E20. doi:10.1002/mus.27373
Comment 2 - Reviewer #2: Content:
For ventilatory support, in G20, 15/34 patients (44%) required Non-Invasive Ventilation (NIV), and one of these underwent a tracheostomy. In G18, in the first year of disease, 10/31 patients (32%) started NIV and none had a tracheostomy in this analyzed period. How does COVID influences the rate of onset of clinical signs? Needs justification for this conclusion.
Reply to reviewer: the difference in terms of ventilatory support is not statistically significant; however, in consideration of a global worsening of all outcomes (as shown in Table n. 2), we believe it may be useful to underline this difference. Unfortunately we do not have a certain explanation of the problem: it could be linked to reduced physical activity and consequently to less training, which permits a more rapid worsening, or to reduced physiotherapy sessions (including ventilatory) linked to the impossibility of access to the hospital.
Comment 3 - Reviewer #2: Content:
Does ALS groups of G20 and G18 were taking any medications for depression before diagnosing the ALS itself? If yes, can you justify this?
Reply to reviewer: we started antidepressant drugs in 13/34 in G20 and in 11/31 in G18 in the first year of disease. Only three other patients for G20 and 2 for G118 were already taking drugs before diagnosis for previously diagnosed depression.
Round 2
Reviewer 2 Report
The authors addressed my concerns.